# Interaction of Depression and Unhealthy Diets on the Risk of Cardiovascular Diseases and All-Cause Mortality in the Chinese Population: A PURE Cohort Substudy

**DOI:** 10.3390/nu14235172

**Published:** 2022-12-05

**Authors:** Xinyue Lang, Zhiguang Liu, Shofiqul Islam, Guoliang Han, Sumathy Rangarajan, Lap Ah Tse, Maha Mushtaha, Junying Wang, Lihua Hu, Deren Qiang, Yingxuan Zhu, Salim Yusuf, Yang Lin, Bo Hu

**Affiliations:** 1Medical Research & Biometrics Center, National Center for Cardiovascular Diseases, The National Clinical Research Center for Cardiovascular Diseases, Fuwai Hospital, Chinese Academy of Medical Sciences & Peking Union Medical College, Beijing 102300, China; 2Department of Pharmacy and Clinical Trial Unit, Beijing Anzhen Hospital, Capital Medical University, Beijing 100029, China; 3Population Health Research Institute, McMaster University and Hamilton Health Sciences, Hamilton, ON L8S 4L8, Canada; 4Division of Occupational and Environmental Health, Jockey Club School of Public Health and Primary Care, The Chinese University of Hong Kong, Hong Kong 999077, China; 5Balingqiao Community Health Service Center, Xinghualing District, Taiyuan 030009, China; 6Nanchang Center for Disease Control and Prevention, Nanchang 330299, China; 7Wujin District Center for Disease Control and Prevention, Changzhou 213022, China

**Keywords:** depression, diet, cardiovascular diseases, all-cause mortality

## Abstract

This study aimed to identify the interaction of depression and diets on cardiovascular diseases (CVD) incident and death in China and key subpopulations. We included 40,925 participants from the Prospective Urban Rural Epidemiology (PURE)-China cohort which recruited participants aged 35–70 years from 45 urban and 70 rural communities. Depression was measured by the adapted Short-Form (CIDI-SF). The unhealthy diet was considered when the score of Alternative Healthy Eating Index was below the lowest tertile. The primary outcome was a composite outcome of incident CVD and all-cause mortality. Cox frailty models were used to examine the associations. During a median follow-up of 11.9 years (IQR: 9.6–12.6 years), depression significantly increased the risk of the composite outcome (HR = 2.00; 95% CI, 1.16–3.27), major CVD (HR = 1.82; 95% CI, 1.48–2.23), and all-cause mortality (HR = 2.21; 95% CI, 1.51–3.24) for the unhealthy diet group, but not for the healthy diet group. The interaction between depression and diet for the composite outcome was statistically significant (RERI = 1.19; 95% CI, 0.66–1.72; AP = 0.42, 95% CI, 0.27–0.61; SI = 3.30, 95% CI, 1.42–7.66; multiplicative-scale = 1.74 95% CI, 1.27–2.39), even in the subgroup and sensitivity analyses. In addition, the intake of vegetable and polyunsaturated fatty acids contributed most to the interaction of diets and depression. Depressive participants should focus on healthy diets, especially vegetables and polyunsaturated fatty acids, to avoid premature death and CVD.

## 1. Introduction

The global epidemic of cardiovascular diseases (CVDs), including in China, is largely attributable to several modifiable risk factors associated with lifestyle and psychological risk factors [1]. Depression as an important psychological disorder becoming more highly prevalent in the general population, and it has been shown to play an important role in cardiovascular disease onset and progression [2]. Approximately 280 million people suffered from depression worldwide, according to the World Health Organization [3]. 

A positive relationship between depression and mortality was consistently reported by cohorts [4,5,6,7,8]. Results from several meta-analyses indicated that depression increased the incidence risk of cardiovascular diseases (CVDs) [9,10,11,12,13,14,15,16,17], such as acute myocardial infarction (AMI) [18] and ischemic heart disease (IHD) [19]. While a wide variation of depression prevalence was reported amongst high-, middle-, and low-income countries [5], the prevalence of depression in the Chinese general population has been less studied. CKB and DFTJ reported a large variation from 0.64% to 17.96% [4]. Moreover, the association of depression with the risk of CVD occurrence and all-cause mortality among the Chinese has yet to be characterized.

An unhealthy diet is another important modifiable lifestyle risk factor characterized as a lack of vegetables, fruits, nuts, soy, and unsaturated fats [20]. The Alternative Healthy Eating Index (AHEI) is a validated dietary index proposed by McCullough et al. [21], which was used to assess the status of unhealthy diets in many epidemiological studies [22]. A cohort study based on 4215 depressive participants indicated that those with a lower score of AHEI had a higher risk of recurrent depression [23]. Another cohort study among 74,930 subjects in the U.S. showed that the 25-percentile higher score of AHEI was associated with a 10% to 20% lower risk of CVDs [24]. More recently, a cross-sectional study demonstrated that an unhealthy diet with inflammatory properties partially mediated the relationship between depressive symptoms and CVD risk, as measured by the Framingham risk score in both men and women [25]. However, the evidence so far indicates that the correlation between dietary factors and depression may be bidirectional, as unhealthy dietary factors may increase the development of depression due to overeating saturated fat and body weight gain [26], whilst the presence of depression may trigger the development of CVD risk factors (e.g., obesity and insulin resistance) via alteration of neuroendocrine hormones and elevation of inflammatory cytokines [27] and, thus, increases the cardiovascular risk.

To date, few large cohort studies considered the effect of depression and unhealthy dietary factors together to investigate the association with CVD risk and all-cause mortality. This study aimed to identify the interaction of depression and diets on the risk of major CVD incidence and all-cause mortality in a large Prospective Urban Rural Epidemiology (PURE) study conducted in China. 

## 2. Materials and Methods

### 2.1. Study Design and Participants

The PURE-China study is a prospective cohort enrolling over 47,000 participants (mainly aged 35–70 years) from 115 communities (45 urban and 70 rural) between 2005 and 2009. The detailed study design has been described previously [28]. For this analysis, we included all outcome events known until April 2021. In addition, we excluded 390 participants (0.8%) who were <35 or >70 years old, 278 participants (0.6%) without follow-up, and 2106 participants (4.4%) who did not answer questions about depression or diet at baseline. To determine the sequence of events, 4232 participants (8.8%) with baseline CVDs and cancer were also excluded. A total of 40,925 individuals were included in our analysis (Appendix A).

### 2.2. Exposure Measurement

Baseline information was measured by trained field researchers using standardized questionnaires. Anthropometrics and fasting blood samples were also collected and recorded. The depression symptoms were measured by an adapted Short-Form Composite International Diagnostic Interview (CIDI-SF) for major depressive disorders [29], which has been used previously in large Chinese cohort studies [4,19]. Participants were first asked whether they felt sad, blue, or depressed for two weeks or longer in the past year. If the answer was “yes”, seven additional questions were asked, including whether they had loss of interest, tired feelings, weight change, difficulty sleeping, difficulty concentrating, thoughts of death, and feeling down. Participants with 3 or more of these 7 symptoms were considered to be depressed [30]. Diet was measured using country-specific food frequency dietary questionnaires (FFQ) and were scored according to the Alternative Healthy Eating Index [21]. The AHEI included 7 components, with the total score ranging from 2.5 to 70. Higher eating scores represent healthier eating patterns (detailed information is provided in the Appendix A). We used the tertiles of the AHEI score to classify the diet group [31], and the scores below the lowest tertiles (T1) were considered as the unhealthy diet group, while the other was considered as the healthy group.

### 2.3. Follow-Up and Outcomes

The primary outcomes were a composite outcome, defined as either the incidence of major CVD (including death from a cardiovascular cause, non-fatal myocardial infarction (MI), stroke, and heart failure) or all-cause mortality. Secondary outcomes were the incidence of major CVD and all-cause mortality. The specific definitions of the events were described previously [5]. 

Follow-up was performed every three years from January 2008 to April 2021. Mortality and major CVD events were recorded mainly from household interviews, medical records, death certificates, and home visits and were adjudicated centrally by trained physicians using standardized definitions [32]. 

### 2.4. Statistical Analysis

Baseline characteristics were described using means with standard deviations for continuous variables and frequencies with percentages for categorical variables. Event rate per 1000 person-years and its 95% CI for all outcomes were compared in people with versus without depression. Due to the heterogeneity among centers, the center number was considered a random effect in the model. Cox proportional hazards models assuming shared frailty were used to assess the associations between depression and mortality. The proportional hazards assumption was also checked using a log-log survival plot. Due to the competing risks caused by non-cardiovascular deaths, for CVD incidence, we used the frailty-based competing risks model [33,34] which extended the Fine–Gray proportional hazards model for clustered data. In Model 1, we adjusted for baseline age, gender, and center (as the random effect). Model 2 was further adjusted for urban/rural regions, physical activities, education level, wealth index, social isolation index (based on the Modified Social Network Index [35]), and self-reported disabilities score. In Model 3, we further adjusted baseline confounders, including current smoking, alcohol use, hypertension, diabetes, dyslipidemia, central obesity (waist-to-hip ratio), and the use of statins. More detailed information about the classification was described in the previous study [5]. The interaction of depression and diet was measured by relative excess risks due to interaction (RERI), attributable proportion due to interaction (AP), the synergy index (SI), multiplicative-scale interaction, and their 95% confidence intervals (CIs) [36,37] (Detailed information is in Appendix A). Restricted cubic splines were then used to explore the dose–response relationship of the association between CIDI-SF score (as a continuous variable) and the outcomes with the covariates in Model 3. Subgroup analyses of key populations (grouped by age, gender, and urban/rural regions) and the AHEI components (vegetable score, fruit score, nuts, and soy protein score, ratio of white to red meat score, cereal fiber score, and ratio of polyunsaturated to saturated fatty acids score (P:S)) were carried out. In addition, each AHEI component was divided into the unhealthy group (the score under T1) and the healthy group (the score over T1). Owing to 86% participants having a full score of trans-fat score, we did not analysis this component. Sensitivity analyses were conducted by excluding events collected in the first 2 years, excluding patients who used anti-depression medicine at baseline or during the follow-ups, patients with diabetes, and patients with heavy alcohol intake (drank > 38% liquor, currently moderate and high). All analyses were conducted using R version 4.0.3. A 2-sided *p* < 0.05 was considered significant. 

## 3. Results

### 3.1. Sample Demographic Characteristics

Of the 47,931 participants recruited for the PURE-China study from 2005 to 2009, 40,925 participants were included in our study. The prevalence of depression in the baseline survey of the PURE-China study was 2.4% (n = 952) for the overall participants. In addition, there was no significant difference in depression prevalence in the healthy diet group (8213 (2.5%)) versus the unhealthy diet group (285 (2.1%)) (Table 1). For the total population, depression was more likely to be reported in individuals who were women (65.9% vs. 57.9% without depression), living in urban areas (50.9% vs. 47.9%), with low relative wealth (40.8% vs. 31.3%), more educated (21.5% vs. 14.2%), socially isolated (17.4% vs. 12.0%), and with more than 2 disabilities (24.2% vs. 9.8%). The distributions of gender, living area, relative wealth, education, social isolation, and disabilities were similar for the healthy diet group and the unhealthy diet group. Compared with the unhealthy diet group, the healthy diet group had more participants who had high relative wealth (13.0% vs. 6.8%) and high education (16.5% vs. 9.8%), but fewer participants were socially isolated (9.7% vs. 17.1%). Participants in the unhealthy group had fewer fruits, nuts, white meat, fiber, polyunsaturated fatty acids (PUFAs), and protein and had more carbohydrates. No significant difference in the AHEI score existed in people with versus without depression. (Appendix A). Detailed information of alcohol use was shown in Appendix A.

### 3.2. Association of Depression with Cardiovascular Diseases and All-Cause Mortality in People with Healthy and Unhealthy Diets

During a median follow-up of 11.9 years (interquartile range (IQR) 9.6–12.6 years), 2066 deaths and 3099 major CVD occurred, comprising 4439 occurrences of the composite outcome. The cause of death was mainly cardiac and circulatory system diseases (694 (33.6%)) and cancer-related (700 (33.9%)), followed by injury (158 (7.6%)) and respiratory diseases (89 (4.3%)). For the total population, event rates for all conditions were higher in people with depression compared with the people without depression. In addition, for the unhealthy diet group, the difference in the event rate of depressed and not depressed individuals was more obvious, while for the healthy diet group, the event rates of all conditions were similar (Table 2 and Appendix A). 

Figure 1 provides the results of the adjusted hazard ratio (HR) models for depression with the primary outcomes and the secondary outcomes of the total population, the healthy diet group, and the unhealthy diet group. For the total population, after adjusting for age, sex, and center (as a random effect), depression was associated with the composite outcome and all-cause mortality. In the fully adjusted models, the associations still existed, and the HRs for the primary outcomes increased by 38% in people with depression. In addition, for the secondary outcomes, the HRs increased by 40% and 44%, respectively, (major CVD: HR = 1.40, 95% CI, 1.09–1.81; all-cause mortality: HR = 1.441; 95% CI, 1.11–1.85). As for the healthy diet group, depression did not increase the risk of any outcome. However, for the unhealthy diet group, the impacts of depression became larger for the composite outcome (HR = 2.00; 95% CI, 1.49–2.68), major CVD (HR = 1.82; 95% CI, 1.48–2.23), and all-cause mortality (HR = 2.21; 95% CI, 1.51–3.24). 

Restricted cubic splines indicated a linear relationship between the number of depressive symptoms measured by the CIDI-SF score and each of the three outcomes. For the healthy diet group, the associations were not significant for each outcome. However, for the unhealthy diet group, the risk of all-cause mortality, major CVD, and the composite outcome was more obviously increased with the increased number of depressive symptoms (Appendix A).

### 3.3. The Additive and Multiplicative Interaction of Depression and Diets for Cardiovascular Diseases and All-Cause Mortality

Interaction of depression symptoms and diet was observed for the outcomes. More specifically, for the composite outcomes, after adding the interaction term of depression and diet group, the independent effect of depression and unhealthy diet was mild, HR = 1.42 (95% CI, 1.05–1.93), and HR = 1.10 (95% CI, 1.02–1.18), respectively. However, the synergy effect of depression and unhealthy diet was significant, HR = 2.71 (95% CI, 2.26–3.26), RERI = 1.19 (95% CI, 0.66–1.72, *p* < 0.001), AP = 0.42 (95% CI, 0.27–0.61, *p* < 0.001), SI = 3.30 (95% CI, 1.42–7.66, *p* < 0.001), and multiplicative interaction = 1.74 (95% CI, 1.27–2.39, *p* = 0.001) (Table 3).

For major CVD, the independent effect of depression was significant (HR = 1.27, 95% CI, 1.01–1.77), while unhealthy diet was not (HR = 1.06, 95% CI, 0.98–1.15). However, the synergistic effect of depression and unhealthy diet was significant, HR = 2.17 (95% CI, 1.45–3.23), RERI = 0.84 (95% CI, 0.40–1.28, *p* < 0.001), AP = 0.39 (95% CI, 0.21–0.57, *p* < 0.001), SI = 3.58 (95% CI, 1.07–11.91, *p* < 0.001), and multiplicative interaction = 1.61 (95% CI, 1.19–2.18, *p* = 0.002) (Table 3). 

For all-cause mortality, the independent effect of depression was significant (HR = 1.38, 95% CI, 1.06–1.80), while unhealthy diet was not (HR = 1.04, 95% CI, 0.94–1.14). However, the synergy effect of depression and unhealthy diet was significant, HR = 2.90 (95% CI, 1.86–4.53), RERI = 1.48 (95% CI, 0.48–3.04), *p* < 0.001), AP = 0.51 (95% CI, 0.16–0.66, *p* < 0.001), SI = 4.57 (95% CI, 1.31–15.91, *p* < 0.001), and multiplicative interaction = 2.03 (95% CI, 1.28–3.21, *p* = 0.003) (Table 3). Subgroup analyses in key subpopulations (participants aged 45–55 years and 55–70 years; male and female; living in urban and rural regions) also indicated the additive and multiplicative interaction of depression and unhealthy diets on the composite outcome, except for young age individuals (35–45 years). Additional sensitivity analyses were conducted by excluding 313 patients who had events in the first 2 years, 30 patients who used anti-depression medicine at baseline or during the follow-ups, 2927 patients with diabetes, and 454 patients with heavy alcohol intake. These analyses indicated consistent results (Table 3).

### 3.4. Subgroup Analyses of the AHEI Components 

Subgroup analyses of the AHEI components showed the interaction of depression symptoms with vegetable score (*p* for interaction < 0.05) and P:S score (*p* for interaction < 0.05) for the composite outcome and major CVD (Figure 2). In addition, for vegetable score and P:S score≥ T1, depression did not increase the risk of any outcome. However, for vegetable score and P:S score < T1, the impacts of depression became significant for the composite outcome (vegetable score < T1, HR = 1.73, 95% CI, 1.33–2.25; P:S score < T1, HR = 1.67, 95% CI, 1.26–2.23) and major CVD (vegetable score < T1, HR = 1.90, 95% CI, 1.38–2.63; P:S score < T1, HR = 1.85, 95% CI, 1.33–2.56).

## 4. Discussion

In this cohort study, we prospectively observed that adults with depression had more risk of CVD incidence and all-cause mortality. In addition, a healthy diet could help decrease the risk while an unhealthy diet significantly increased the risk of the composite outcome, the morbidity of major CVD, and death. The additive and multiplicative interaction between depression and diet was significant for the outcomes, the same in the key subpopulations, except for participants aged 35–45 years. People with an unhealthy diet tend to intake more carbohydrates and less protein, fruits, nuts, white meat, fiber, and polyunsaturated fatty acids. The intake of vegetable and polyunsaturated fatty acids contributed more to the interaction of diet and depression for the outcomes.

The association of depression with all-cause mortality (HR = 1.44, 95% CI, 1.11–1.85) was similar to a meta-analysis [39] (RR = 1.34, 95% CI, 1.27–1.42) and CKB study [4] (1.32 (95% CI, 1.20–1.46)) but slightly higher than that of the DFTJ study [4] (1.17 (95% CI, 1.06–1.29)). As for the relationship of depression with CVD, our result (HR = 1.40, 95% CI, 1.09–1.81) was similar with CKB (HR = 1.33, 95% CI, 1.15–1.53) [19] and CHARLS study (HR = 1.39, 95% CI, 1.22–1.58) [40]. Although using the same depression scale, the depression prevalence of our PURE-China study (2.4%) was much higher than that of the CKB cohort study (0.64%) but much lower than that of DFTJ participants (17.96%). However, our prevalence was very similar to that of the WHO’s global prevalence (5.0%) for adults worldwide [3]. DFTJ cohort included 27,009 Dongfeng Motor workers, mainly retired, with an average age of 63.6. The CKB cohort selected 512,712 participants, but only from five urban and five rural areas. PURE-China included 45 urban and 70 rural areas of the Chinese population and, therefore, had a better representation.

Our study observed differences in the associations of depression with cardiovascular disease and all-cause mortality between healthy and unhealthy diets and emphasized the additive and multiplicative interaction of depression and diet. No previous studies attempted to examine the joint effect of depression and diet together on CVDs/death in the general population. This study took advantage of a large cohort with longer follow-up years to provide a clear picture that the interaction of depression and unhealthy diet provides another 1.17-fold risk of CVDs/death and accounted for 46% of the risk in the joint effect, which is the first endeavor in this area of research and, thus, offers the major merit of this study. From the literature review, there was only one retrospective study [41] that demonstrated that an unhealthy diet (Mediterranean diet score at the lowest tertile) aggravated the adverse impact of depressive symptoms on the 30-day CVD reoccurrence or CVD death in elderly, acute coronary syndrome survivors, while a healthy diet (Mediterranean diet score at the upper tertile) counteracted the impact. The interaction of depressive symptoms and diet for the CVD risk was also indicated by a cross-sectional study [25], which found the inflammatory properties of diet mediated the effect of depressive symptoms.

The possible mechanism of the interaction of depressive disorder and unhealthy diet (especially lack of vegetables and polyunsaturated fatty acids) on increasing the additional risk of CVDs and all-cause mortality could be explained by the over activation of platelets and endothelial dysfunction of the vessel, which may lead to an increasing level of circulating inflammatory cytokines, and these adverse biological processes eventually promote atherosclerosis, which is one of the major risk factors of CVDs [27,42,43]. This potential biological mechanism was further supported by clinical trials that depressive disorder or lack of omega-3 fatty acids could increase corticotropin-releasing hormones [44,45], causing hypothalamic–pituitary–adrenal axis dysfunction, and this process is involved in the development of the metabolic syndrome and cardiovascular diseases.

Meta-analyses have found a significant association between depressive symptom reduction and eating a healthy diet [46,47]. In addition, a randomized clinical trial found that adults with elevated depression symptoms under the diet intervention (i.e., a healthy diet) showed significantly reduced symptoms of depression compared with the habitual diet control group [48]. Therefore, we speculate that eating a healthy diet could improve depressive symptoms and further reduce the risk of CVDs and death, but this needs to be verified by further research.

Our study firstly discovered the additive and multiplicative interaction of depression and diet on all-cause mortality and cardiovascular events and was based on a large-scale Chinese cohort study. Quality-controlled baseline and follow-up information collection were conducted. In addition, we considered the use of anti-depressants medicine (only 3.2%), and the treatment rate was similar to that of the previous study [49]. However, some limitations should not be ignored. First, we analyzed the depression and diet data only at baseline. Depression and unhealthy diet might influence each other. Thus, mental health and dietary changes might have occurred during the follow-up period, which probably would have weakened the observed associations. Second, the depression was measured by a simplified scale, and the potential misclassifications might exist. However, a single globally validated screening method for depression is lacking, and the use of a complex scale [50] in combination with a conversation to accurately diagnose the depression disorder is not feasible in large cohort studies of general populations. The misclassifications of depression might weaken the association between depression and the events, and, therefore, the real influence of depression could be larger than the results we have observed. Third, the record for food thought FFQs was not based on a photograph, but it was suitable for classifying individuals into intake categories and was the most common method for assessing food intake in cohort studies. Last, like other observational studies, the effect of residual confounding should not be ignored. However, we adjusted the possible confounding as much as possible; therefore, the impact will be small.

## 5. Conclusions

Depression and unhealthy diets had additive and multiplicative interaction on the occurrence of all-cause mortality and CVD, the same in the key subpopulations (participants aged 45–55 years and 55–70 years; male and female; living in urban and rural regions). If the health-related sustainable development goals are to be achieved, the government should raise awareness of the physical health risks associated with depression and encourage people with depressive symptoms to develop a healthy diet and increase their intake of vegetables and unsaturated fatty acids. Further studies of dietary intervention on depression patients are still needed.

## Figures and Tables

**Figure 1 nutrients-14-05172-f001:**
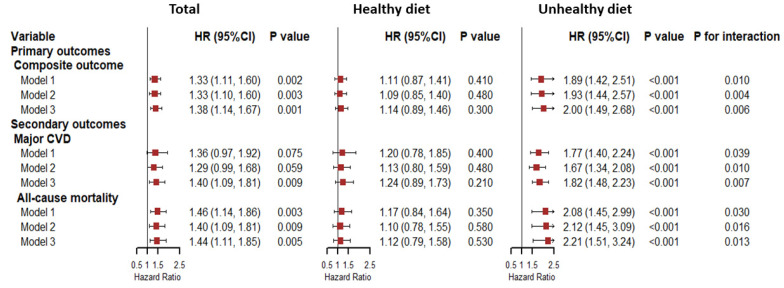
Forest plots for the associations between depression and the outcomes by diet. In Model 1, we adjusted for baseline age, gender, and center (as the random effect). Model 2 was further adjusted for urban/rural regions, physical activities, education level, wealth index, social isolation index, and self-reported disabilities score. In Model 3, we further adjusted baseline confounders, including current smoking, alcohol use, hypertension, diabetes, dyslipidemia, central obesity, and use of statins. Abbreviations: HR, hazard ratio; CI, confidence interval; CVD, cardiovascular disease.

**Figure 2 nutrients-14-05172-f002:**
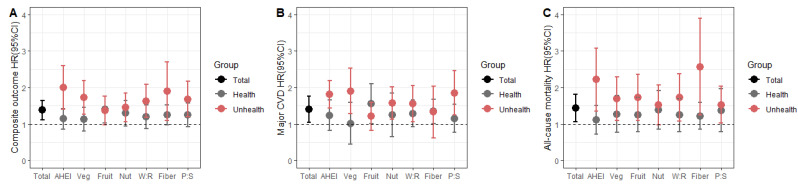
Subgroup analysis of the associations between depression and the outcomes by AHEI component. (**A**) Composite outcome, (**B**) Major CVD, and (**C**) All-cause mortality. ‘Health’ was considered that the AHEI component score (vegetable score, fruit score, nuts, and soy protein score, ratio of white to red meat score, cereal fiber score, and ratio of polyunsaturated to saturated fatty acids score (P:S)) over the lowest tertiles. In addition, ‘Unhealth’ was considered the AHEI component score below the lowest tertiles. The models were adjusted for age, gender, urban/rural regions, physical activities, education level, wealth index, social isolation index, self-reported disabilities score, current smoking, alcohol use, hypertension, diabetes, dyslipidemia, central obesity, use of statins, and center. Abbreviations: AHEI, Alternative Healthy Eating Index; CI, confidence interval; CVD, cardiovascular disease; HR, hazard ratio; P:S, ratio of polyunsaturated to saturated fatty acids score; W:R, ratio of white to red meat score.

**Table 1 nutrients-14-05172-t001:** Baseline information of participants grouped by depression and diet.

Variables	Total		Healthy Diet		Unhealthy Diet	
	without Depression (N = 39,973)	with Depression (N = 952)	without Depression (N = 26,635)	with Depression (N = 667)	without Depression (N = 13,338)	with Depression (N = 285)
Age mean (SD), years	50.6 ± 9.6	48.7 ± 8.8	50.6 ± 9.6	48.5 ± 8.8	50.5 ± 9.6	49.2 ± 8.8
Male, (%)	16,822 (42.1)	325 (34.1)	11,190 (42)	232 (34.8)	5632 (42.2)	93 (32.6)
Female, (%)	23,151 (57.9)	627 (65.9)	15,445 (58.0)	435 (65.2)	7706 (57.8)	192 (67.4)
Living area						
Urban	19,135 (47.9)	485 (50.9)	13,697 (51.4)	360 (54.0)	5438 (40.8)	125 (43.9)
Rural	20,838 (52.1)	467 (49.1)	12,938 (48.6)	307 (46.0)	7900 (59.2)	160 (56.1)
Relative wealth ^a^						
Low	12,357 (31.3)	380 (40.8)	7589 (28.8)	248 (37.9)	4768 (36.4)	132 (47.7)
Moderate	22,716 (57.6)	435 (46.7)	15,266 (58.0)	313 (47.9)	7450 (56.8)	122 (44.0)
High	4348 (11.0)	116 (12.5)	3451 (13.1)	93 (14.2)	897 (6.8)	23 (8.3)
Education ^b^						
Low	13,344 (33.5)	337 (35.5)	8034 (30.2)	193 (29.0)	5310 (40.0)	144 (50.7)
Moderate	20,873 (52.4)	409 (43.1)	14,188 (53.4)	314 (47.1)	6685 (50.3)	95 (33.5)
High	5651 (14.2)	204 (21.5)	4356 (16.4)	159 (23.9)	1295 (9.7)	45 (15.8)
Current Smoker	9224 (23.3)	191 (20.2)	6251 (23.6)	131 (19.8)	2973 (22.6)	60 (21.2)
Alcohol use ^c^						
Never	29,922 (76.1)	692 (73.8)	19,881 (75.7)	486 (73.7)	10,041 (76.8)	206 (73.8)
Former	1095 (2.8)	60 (6.4)	714 (2.7)	48 (7.3)	381 (2.9)	12 (4.3)
Current Low	6718 (17.1)	140 (14.9)	4588 (17.5)	98 (14.9)	2130 (16.3)	42 (15.1)
Current Moderate	1326 (3.4)	38 (4.1)	897 (3.4)	22 (3.3)	429 (3.3)	16 (5.7)
Current High	281 (0.7)	8 (0.9)	185 (0.7)	5 (0.8)	96 (0.7)	3 (1.1)
Low trust in others ^d^	1166 (2.9)	28 (3.0)	687 (2.6)	17 (2.6)	479 (3.6)	11 (3.9)
Socially isolated ^e^	4802 (12.0)	166 (17.4)	2524 (9.5)	113 (16.9)	2278 (17.1)	53 (18.6)
Physical activity ^f^						
Low	6098 (15.5)	132 (14.4)	3410 (13.0)	90 (14.0)	2688 (20.6)	42 (15.3)
Moderate	16,620 (42.3)	324 (35.3)	11,294 (43.1)	229 (35.7)	5326 (40.7)	95 (34.5)
High	16,556 (42.2)	461 (50.3)	11,494 (43.9)	323 (50.3)	5062 (38.7)	138 (50.2)
Disabilities ^g^						
0	32,294 (80.9)	595 (62.5)	22,200 (83.4)	419 (62.8)	10,094 (75.7)	176 (61.8)
1	3744 (9.4)	127 (13.3)	2121 (8.0)	89 (13.3)	1623 (12.2)	38 (13.3)
≥2	3896 (9.8)	230 (24.2)	2287 (8.6)	159 (23.8)	1609 (12.1)	71 (24.9)
Hypertension ^h^ (%)	15,838 (40.1)	302 (32.1)	10,822 (41.2)	209 (31.6)	5016 (38.0)	93 (33.2)
Diabetes mellitus ^i^ (%)	2857 (7.1)	70 (7.4)	1928 (7.2)	50 (7.5)	929 (7.0)	20 (7.0)
Dyslipidemia ^j^ (%)	18,469 (46.2)	368 (38.7)	12,454 (46.8)	263 (39.4)	6015 (45.1)	105 (36.8)
Abdominal obesity ^k^	16,380 (41.6)	352 (37.4)	10,878 (41.5)	241 (36.6)	5502 (41.9)	111 (39.4)
Use of statins	366 (0.9)	11 (1.2)	238 (0.9)	7 (1.0)	128 (1.0)	4 (1.4)

^a^ Relative wealth: Weighted according to housing characteristics and assets; divide family wealth from the poorest to the richest into thirds [38]. ^b^ Education: low, primary education level or less; moderate, secondary school education; high, college/trade school/university education. ^c^ Alcohol use: current low, alcohol drinks < 1 time/day; currently moderate, alcohol drinks 1–3 times/day (men)/1–2 times/day (women); currently high, alcohol drinks > 3 times/day (men)/>2 times/day (women). ^d^ Low trust: the belief that people were generally not honest and helpful and that doing nice things for someone would be unlikely to be reciprocated. ^e^ Social isolation: a score of 4 of 5 on the Modified Social Network Index [35]. ^f^ Physical activity: low (<600 metabolic equivalents (MET) × min/week or <150 min/week of moderate-intensity physical activity), moderate (600–3000 MET × min or 150–750 min/week), or high (>3000 MET × min or >750 min/week). ^g^ Disabilities: 0, 1, or 2 of difficulty grasping, walking, bending, reading, seeing people, speaking/hearing, and using walking aids. ^h^ Hypertension: systolic blood pressure > 140 mm Hg, diastolic blood pressure > 100 mm Hg/diagnosed with hypertension or taking hypertension medication. ^i^ Diabetes: fasting glucose levels, 126.13 mg/dL (to convert to millimole per liter, multiply by 0.0555) or previously diagnosed diabetes or use of glucose-lowering medications. ^j^ Dyslipidemia: total cholesterol ≥ 6.2 mmol/L, or low-density lipoprotein cholesterol ≥ 4.1 mmol/L, or triglyceride ≥ 2.3 mmol/L. ^k^ Abdominal obesity: waist to hip ratio, 0.9 (men)/0.85 (women).

**Table 2 nutrients-14-05172-t002:** Event rates grouped by depression and diet.

Variable	Total		Healthy Diet		Unhealthy Diet	
	without Depression (N = 39,973)	with Depression (N = 952)	without Depression (N = 26,635)	with Depression (N = 667)	without Depression (N = 13,338)	with Depression (N = 285)
Primary outcomes						
Composite outcome						
Events, N (%)	4317 (10.8)	122 (12.8)	2797 (10.5)	70 (10.5)	1520 (11.4)	52 (18.2)
Event rate/1000 person-years (95% CI)	10.2 (9.9, 10.5)	12.2 (10.0, 14.3)	9.8 (9.5, 10.2)	10.2 (7.8, 12.6)	11.1 (10.5, 11.6)	19.0 (13.9, 24.2)
Secondary outcomes						
Major CVD						
Events, N (%)	3022 (7.6)	77 (8.1)	1989 (7.5)	47 (7.0)	1033 (7.7)	30 (10.5)
Event rate/1000 person-years (95% CI)	7.2 (6.9, 7.4)	7.7 (6.0, 9.4)	7.0 (6.7, 7.3)	6.9 (4.9, 8.8)	7.5 (7.1, 8.0)	11.0 (7.1, 14.9)
All-cause mortality						
Events, N (%)	1998 (5.0)	68 (7.1)	1268 (4.8)	36 (5.4)	730 (5.5)	32 (11.2)
Event rate/1000 person-years (95% CI)	4.6 (4.4, 4.8)	6.6 (5.0, 8.2)	4.4 (4.1, 4.6)	5.1 (3.5, 6.8)	5.2 (4.8, 5.6)	11.3 (7.4, 15.3)

A composite outcome was defined as the first of either a major cardiovascular event or death. Abbreviations: CVD, cardiovascular disease; CI, confidence interval.

**Table 3 nutrients-14-05172-t003:** Interaction of depression and unhealthy diet for the outcomes.

	Risk of the Outcomes HR (95% CI) ^b^				Additive Interaction (95% CI)			Multiplicative Interaction ^c^ (95% CI)
	No Depression or Unhealthy Diet	Depression	Unhealthy Diet	Depression and Unhealthy Diet	RERI	AP	SI	
Composite outcome ^a^	Ref.	1.42 (1.05, 1.93) *p* = 0.023	1.10 (1.02, 1.18) *p* = 0.011	2.71 (2.26, 3.26) *p* < 0.001	1.19 (0.66, 1.72) *p* < 0.001	0.44 (0.27, 0.61) *p* < 0.001	3.30 (1.42, 7.66) *p* < 0.001	1.74 (1.27, 2.39) *p* = 0.001
Major CVD	Ref.	1.27 (1.01, 1.77) *p* = 0.046	1.06 (0.98, 1.15) *p* = 0.165	2.17 (1.45, 3.23) *p* < 0.001	0.84 (0.40, 1.28) *p* < 0.001	0.39 (0.21, 0.57) *p* < 0.001	3.58 (1.07, 11.91) *p* < 0.001	1.61 (1.19, 2.18) *p* = 0.002
All-cause mortality	Ref.	1.38 (1.06, 1.80) *p* = 0.017	1.04 (0.94, 1.14) *p* = 0.474	2.90 (1.86, 4.53) *p* < 0.001	1.48 (0.48, 3.04) *p* < 0.001	0.51 (0.16, 0.66) *p* < 0.001	4.57 (1.31, 15.91) *p* < 0.001	2.03 (1.28, 3.21) *p* = 0.003
**Subgroup analysis for the Composite outcome**								
Age								
35–45	Ref.	1.73 (1.08, 2.77) *p* = 0.024	1.13 (1.01, 1.27) *p* = 0.032	2.54 (0.99, 6.51) *p* = 0.051	0.68 (−1.26, 2.63) *p* = 0.246	0.27 (−0.33, 0.87) *p* = 0.190	1.79 (0.41, 7.89) *p* = 0.015	1.30 (0.57, 2.96) *p* = 0.535
45–55	Ref.	1.54 (1.02, 2.31) *p* = 0.039	1.10 (1.00, 1.20) *p* = 0.051	3.06 (1.72, 5.45) *p* < 0.001	1.42 (0.34, 2.50) *p* = 0.005	0.47 (0.23, 0.70) *p* < 0.001	3.24 (1.21, 8.70) *p* < 0.001	1.81 (1.14, 2.89) *p* = 0.012
55–70	Ref.	1.23 (0.89, 1.69) *p* = 0.203	1.09 (0.98, 1.22) *p* = 0.100	2.46 (1.50, 4.04) *p* < 0.001	1.14 (0.43, 1.85) *p* = 0.001	0.46 (0.27, 0.66) *p* < 0.001	4.53 (1.25, 16.41) *p* < 0.001	1.83 (1.25, 2.69) *p* = 0.002
Gender								
Male	Ref.	1.56 (1.03, 2.37) *p* = 0.037	1.07 (1.00, 1.15) *p* = 0.057	3.28 (2.34, 4.59) *p* < 0.001	1.64 (0.41, 2.87) *p* = 0.004	0.50 (0.25, 0.76) *p* < 0.001	3.61 (1.16, 11.23) *p* < 0.001	1.96 (1.16, 3.31) *p* = 0.012
Female	Ref.	1.34 (0.95, 1.87) *p* = 0.091	1.13 (1.03, 1.24) *p* = 0.010	2.50 (1.59, 3.94) *p* < 0.001	1.04 (0.53, 1.55) *p* < 0.001	0.41 (0.25, 0.58) *p* < 0.001	3.23 (1.30, 8.05) *p* < 0.001	1.66 (1.21, 2.28) *p* = 0.002
Region								
Urban	Ref.	1.06 (1.00 1.57) *p* = 0.050	1.15 (1.00, 1.31) *p* = 0.044	2.34 (1.26, 4.34) *p* = 0.007	1.13 (0.15, 2.11) *p* = 0.012	0.48 (0.21, 0.76) *p* < 0.001	6.52 (1.65, 65.64) *p* < 0.001	1.93 (1.09, 3.39) *p* = 0.023
Rural	Ref.	1.66 (1.15, 2.39) *p* = 0.007	1.07 (0.98, 1.16) *p* = 0.147	2.82 (2.17, 3.68) *p* < 0.001	1.10 (0.39, 1.82) *p* = 0.001	0.39 (0.18, 0.60) *p* < 0.001	2.53 (1.13, 5.69) *p* < 0.001	1.60 (1.11, 2.31) *p* = 0.011
**Sensitivity analysis for the composite outcome**								
Omitting events in first 2 years	Ref.	1.10 (1.02, 1.18) *p* = 0.009	1.35 (1.01, 1.81) *p* = 0.041	2.50 (1.76, 3.56) *p* < 0.001	1.05 (0.58, 1.51) *p* < 0.001	0.42 (0.27, 0.56) *p* < 0.001	3.31 (1.48, 7.39) *p* < 0.001	1.68 (1.29, 2.19) *p* < 0.001
Omitting those using anti-depression drugs	Ref.	1.10 (1.02, 1.18) *p* = 0.012	1.42 (1.05, 1.93) *p* = 0.023	2.71 (2.26, 3.25) *p* < 0.001	1.19 (0.66, 1.72) *p* < 0.001	0.44 (0.27, 0.61) *p* < 0.001	3.30 (1.42, 7.68) *p* < 0.001	1.74 (1.27, 2.39) *p* = 0.001
Omitting those with diabetes	Ref.	1.18 (1.07, 1.31) *p* = 0.002	1.05 (0.71, 1.55) *p* = 0.808	2.20 (2.07, 2.35) *p* < 0.001	0.97 (0.64, 1.30) *p* < 0.001	0.44 (0.29, 0.59) *p* < 0.001	5.17 (0.86, 31.04) *p* < 0.001	1.77 (1.29, 2.44) *p* < 0.001
Omitting those with heavy alcohol intake	Ref.	1.10 (1.02, 1.18) *p* = 0.015	1.43 (1.06, 1.92) *p* = 0.018	2.76 (2.24, 3.40) *p* < 0.001	1.24 (0.69, 1.78) *p* < 0.001	0.45 (0.29, 0.61) *p* < 0.001	3.37 (1.48, 7.65) *p* < 0.001	1.76 (1.29, 2.42) *p* < 0.001

^a^ Composite outcome was defined as the first of either a major cardiovascular event or death. ^b^ The models were adjusted for the interaction term of depression and diet group, baseline age, gender, urban/rural regions, physical activities, education level, wealth index, social isolation index, self-reported disabilities score, current smoking, alcohol use, hypertension, diabetes, dyslipidemia, central obesity, and use of statins, with center as the random effect. Age, sex, or urban/rural regions was not considered as a covariate in each subgroup analysis. ^c^ Null hypothesis for each interaction was RERI = 0, AP = 0, SI = 1, and multiplicative interaction = 1. Abbreviations: CVD, cardiovascular disease; HR, hazard ratio; CI, confidence interval; RERI, relative excess risks; AP, attributable proportion; SI, synergy index.

## Data Availability

The PURE study is ongoing and data are not currently publicly available. Formal collaboration with other groups with similar data for expanded and pooled analyses will be considered.

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
