# Peer review of "Interaction of Depression and Unhealthy Diets on the Risk of Cardiovascular Diseases and All-Cause Mortality in the Chinese Population: A PURE Cohort Substudy"

_nutrients, 2022, doi:10.3390/nu14235172_

Round 1
Reviewer 1 Report
The study is devoted to the problem of identifying the effect of both depression and diets on the risk of major cardiovascular disease (CVD) incidence and all-cause mortality rate in Chinese population.
It is known that diet and nutrition itself (especially the fact of an increasing intake of cholesterol and a lack of vegetables, fruits, nuts, and unsaturated fat) have a great impact on the organism and might be linked to potential development of diseases.
It is quite interesting to analyze the association between the diet, psychological risk factors and cardiovascular diseases incidence.
The conducted study helps to understand that changing the diet might make it possible to diminish depression effect on CVD development.
The cohort in the research was chosen carefully, as age of individuals, cancer and CVD diseases, responses of the participants to the questionnaire were considered.
The population which had been included in the research is rather large, so the obtained data should be informative and precise. Participants from rural and urban territories were taken into account which is also quite interesting, because of the peculiarities of their lifestyles.
The methods are clearly described, so it is possible to understand the experiment.
Validated dietary index was used to assess the status of unhealthy diet which makes the obtained data even more valuable.
Recommendations in diet scheme are provided to help keep humans health in case of depression development.
The following comments do not diminish the value of the Research:
Line 83, 84 Why the time period from 2005 to 2009 was chosen?
Line 109 ‘The primary outcomes were a composite outcome, defined as either the incidence of major CVD (including deaths from a cardiovascular cause, non-fatal myocardial infarction (MI), stroke, and heart failure) or all-cause mortality’. Why the decision to combine the first of either a major cardiovascular event or death was made? Probably it also would be interesting to take into account the data on the incidence of a major cardiovascular event and death separately?
Table 1 Were men included in the research? Why was not the baseline information about men included in the Table 1? The data in the Table 1, except the ‘Female’ line, involve the participants of both sexes, right? Probably it should be noted in the table description?
Figure 1 The information is presented only for depressed people of both sexes, is it correct?
Lines 233, 234 What does that mean ‘the HRs for the primary outcomes increased by 38% in people with 3 or more depression’? Would it probably might be explained a bit.
Line 234 ‘the HRs increased by 40% and 44% separately’ – probably it would be better to add an information about the data according to which the changes occurred?
Figure 2. ‘Health’ – is for healthy diet and ‘Unhealth’ - is for unhealthy diet, is it correct? Probably it would be better to give the full name of the groups in the figure description.
It would be better to correct the References description according to the requirements published on the website of the Journal.
Author Response
Dear reviewer,
Thanks very much for your thoughtful comments. Changed parts are highlighted in red font in the text. The replies are the following.
Comment 1: Line 83, 84 Why the time period from 2005 to 2009 was chosen?
Response: Thank you for your insightful comments. The baseline information of the PURE study China region was collected in 2005 and ended in 2009[Li W, J Hypertens,2016,34(1):39-46]. We included all the participants from China.
Comment 2: Line 109 ‘The primary outcomes were a composite outcome, defined as either the incidence of major CVD (including deaths from a cardiovascular cause, non-fatal myocardial infarction (MI), stroke, and heart failure) or all-cause mortality’. Why the decision to combine the first of either a major cardiovascular event or death was made? Probably it also would be interesting to take into account the data on the incidence of a major cardiovascular event and death separately?
Response: We used a composite outcome because a lot of confounders were included in the Cox regression model, and a certain number of events were required to make the results robust.
The degree of disability impact of these outcomes is similar; therefore, we consider setting a composite outcome, and other PURE studies also set the primary endpoint in this way [Wang C, EHJ,2019,40(20):1620-1629; Li S,JAMA Cardiol,2022]
In addition, we set the incidence of a major cardiovascular event and death as the secondary outcomes, separately. The result was consistent with the primary outcome (Figure 1-2, Table 3).
Comment 3: Table 1 Were men included in the research? Why was not the baseline information about men included in the Table 1? The data in the Table 1, except the ‘Female’ line, involve the participants of both sexes, right? Probably it should be noted in the table description?
Response: Thanks for your advice. We included male in the research and added the male line to Table 1.
Change: Please see the modified one in Table 1 (Line 178).
Comment 4: Figure 1 The information is presented only for depressed people of both sexes, is it correct?
Response: Figure 1 was analyzed in all the included patients (Total), patients with healthy diet and patients with unhealthy diet;
The HR and its 95%CI were obtained by comparing events rates between the depressed group and the non-depressed group.
Comment 5: Lines 233, 234 What does that mean ‘the HRs for the primary outcomes increased by 38% in people with 3 or more depression’? Would it probably might be explained a bit.
Response: Sorry for the mistake. We were going to say people with 3 or more depressive symptoms, since we considered participants with 3 or more of these 7 symptoms as depressed.
We have changed this sentence to “the HRs for the primary outcomes increased by 38% in people with depression”.
Thank you for your careful correction.
Change: Please see the modified one in Line 234 – 236.
Comment 6: Line 234 ‘the HRs increased by 40% and 44% separately’ – probably it would be better to add an information about the data according to which the changes occurred?
Response: Thanks for your advice.
We have changed this sentence into “And for the secondary outcomes, the HRs increased by 40% and 44% separately (Major CVD: HR=1.40, 95% CI, 1.09-1.81; All-cause mortality: HR=1.441; 95% CI, 1.11-1.85).”.
Change: Please see the modified one in Line 237 – 238.
Comment 7: Figure 2. ‘Health’ – is for healthy diet and ‘Unhealth’ - is for unhealthy diet, is it correct? Probably it would be better to give the full name of the groups in the figure description.
Response: Thanks for your advice.
‘Health’ means that the AHEI component (vegetable score, fruit score, Nuts and soy protein score, Ratio of white to red meat score, cereal fiber score, and ratio of polyunsaturated to saturated fatty acids score (P:S)) score over the lowest tertiles. And ‘Unhealth’ means that the AHEI component score under the lowest tertiles.
In order to avoid confusion with the “Healthy diet” and “Unhealthy diet” (defined by the AHEI total score), we have adopted such a description.
We have changed the Figure 2 legend into “A. Composite outcome, B. Major CVD, and C. All-cause mortality. ‘Health’ was considered that the AHEI component (vegetable score, fruit score, Nuts and soy protein score, Ratio of white to red meat score, cereal fiber score, and ratio of polyunsaturated to saturated fatty acids score (P:S)) score over the lowest tertiles. And ‘Unhealth’ was considered the AHEI component score under the lowest tertiles.
The models were adjusted for age, gender, urban/rural regions, physical activities, education level, wealth index, social isolation index, self-reported disabilities score, current smoking, alcohol use, hypertension, diabetes, dyslipidemia, central obesity, use of statins and center.”
Change: Please see the modified one in Line 299– 305.
Comment 8: It would be better to correct the References description according to the requirements published on the website of the Journal.
Response: Thanks for your advice. We have corrected the References' description as ACS style according to the journal requirements.

Reviewer 2 Report
1. Statistical treatments seem appropriate.
2. It would be better to investigate the frequency and type of alcohol in alcohol intake. Some types of alcohol cause dependence, and the possibility of depression cannot be denied. It would be better to also consider the types of alcohol for high-income earners, the types of alcohol for middle-income earners, and the types of alcohol for low-income earners.
3. A correlation was found between depression and unhealthy eating, but I would like to see a description of whether it is better to improve it in that way in the future.
Author Response
Dear reviewer,
Thanks very much for your thoughtful comments. Changed parts are highlighted in red font in the text. The replies are the following.
Comment 1: Statistical treatments seem appropriate.
Response: Thanks for your comment.
Comment 2: It would be better to investigate the frequency and type of alcohol in alcohol intake. Some types of alcohol cause dependence, and the possibility of depression cannot be denied. It would be better to also consider the types of alcohol for high-income earners, the types of alcohol for middle-income earners, and the types of alcohol for low-income earners.
Response: Thanks for your advice. Self-reported alcohol consumption using a standard alcohol consumption frequency questionnaire. Consumption was categorized as former, never, and current. Current consumption is further classified as current low, alcohol drinks <1 time/day; currently moderate, alcohol drinks 1-3 times/day (men)/ 1-2 times/day(women); currently high, alcohol drinks >3 times/day (men)/ >2 times/day (women). We have added the type of alcohol in alcohol intake in supplementary Appendix B: Table S2. Alcohol use information of participants grouped by depression and diet.
The previous study [Gea, A, BMC Med 2013, 11, 192] indicated that the increased risk of depression was greater for people who drink spirits, so we added a sensitivity analysis on alcohol consumption and deleted the people who drank >38% liquor current moderate and currently high. And the result was the same as the primary analysis (Table 3).
Change: Please see the modified one in Table1, Table3, Line 150, Line 286 and Table S2.
Comment 3: A correlation was found between depression and unhealthy eating, but I would like to see a description of whether it is better to improve it in that way in the future.
Response: Thanks for your suggestion. As the observational study aimed to investigate the factors related to the incidence of depression, no intervention was performed among patients who were on unhealthy eating. Nevertheless, in further research, we would consider conducting an analysis or study to assess the links between the improvement of unhealthy eating and the development of depression.
We have added the description of diet intervention and depression in the discussion part. Above all, we also demonstrated this limitation in the updated manuscript.
Change: Please see the modified one in Line 358 – 364, Line 370-373.

Reviewer 3 Report
Under the methods section, please mention which Equator you have used to report your article. Here this is the checklist for Cohort studies:
The Strengthening the Reporting of Observational Studies in Epidemiology (STROBE) Statement: guidelines for reporting observational studies | EQUATOR Network (equator-network.org)
Use it to organise your paper. Also, fill it out and attach it to your paper as the supplmentary file.
You should provide more details on sampling and recruitment of the participants.
Describe each data collection tool in terms of validity/reliability assessments before use, scorping and interpretation.
Conclusion should contain implications for practice and policy making. What should be done based on your findings?
Author Response
Dear reviewer,
Thanks very much for your thoughtful comments. Changed parts are highlighted in red font in the text. The replies are the following.
Comment 1: Under the methods section, please mention which Equator you have used to report your article. Here this is the checklist for Cohort studies:
The Strengthening the Reporting of Observational Studies in Epidemiology (STROBE) Statement: guidelines for reporting observational studies | EQUATOR Network (equator-network.org)
Use it to organize your paper. Also, fill it out and attach it to your paper as the supplementary file.
Response: Thanks for your kind advice. We have organized our paper according to STROBE, and attach it as the supplementary file named “STROBE_checklist_cohort”.
Comment 2: You should provide more details on sampling and recruitment of the participants.
Response: Thanks for your advice. We have added the detail information on sampling and recruitment of the participants in the Supplementary Appendix A2: PURE Study Participant Selection Methodology.
Change: Please see the modified one in Supplementary Appendix A2.
Comment 3: Describe each data collection tool in terms of validity/reliability assessments before use, scorping and interpretation.
Response: Thanks for your advice. We have added the detail information of data collection tool in the Supplementary Appendix A3: Data collection, Supplementary Appendix A4: Standardized Event Definitions in PURE.
Change: Please see the modified one in Supplementary Appendix A3 and A4.
Comment 4: Conclusion should contain implications for practice and policy making. What should be done based on your findings?
Response: Thanks for your kind advice. We have modified the conclusion as the following:
Depression and unhealthy diets had additive and multiplicative interaction on the occurrence of all-cause mortality and CVD, the same in the key subpopulations (participants aged 45-55 years and 55-70 years; male and female; living in urban and rural regions). If the health-related sustainable development goals are to be achieved, the government should raise awareness of the physical health risks associated with depression, and encourage people with depressive symptoms to develop a healthy diet and increase their intake of vegetables and unsaturated fatty acids. Further studies of dietary intervention on depression patients were still needed.
Change: Please see the modified one in Line 387-394.

Round 2
Reviewer 3 Report
Nothing.